# Plasma density limits for hole boring by intense laser pulses

Natsumi Iwata[1], Sadaoki Kojima[1,2], Yasuhiko Sentoku[1], Masayasu Hata[1] & Kunioki Mima[3]

High-power lasers in the relativistic intensity regime with multi-picosecond pulse durations are available in many laboratories around the world. Laser pulses at these intensities reach giga-bar level radiation pressures, which can push the plasma critical surface where laser light is reflected. This process is referred to as the laser hole boring (HB), which is critical for plasma heating, hence essential for laser-based applications. Here we derive the limit density for HB, which is the maximum plasma density the laser can reach, as a function of laser intensity. The time scale for when the laser pulse reaches the limit density is also derived. These theories are confirmed by a series of particle-in-cell simulations. After reaching the limit density, the plasma starts to blowout back toward the laser, and is accompanied by copious superthermal electrons; therefore, the electron energy can be determined by varying the laser pulse length.

[1] Institute of Laser Engineering, Osaka University, 2-6 Yamadaoka, Suita, Osaka 565-0871, Japan. [2] Advanced Research Center for Beam Science, Institute for Chemical Research, Kyoto University, Gokasho, Uji, Kyoto 611-0011, Japan. [3] The Graduate School for the Creation of New Photon Industries, 1955-1 Kurematsu, Nishiku, Hamamatsu, Shizuoka 141-1201, Japan. Correspondence and requests for materials should be addressed to N.I. (email: iwata-n@ile.osaka-u.ac.jp)

With the development of kiloJoule class high-power lasers, intense pulses of laser light in the relativistic intensity level exceeding $10^{18}$ W $\mu$m$^2$ cm$^{-2}$ with picosecond (ps) to multi-ps pulse duration are now available at LFEX[1], NIF-ARC[2], LMJ-PETAL[3], and OMEGA-EP[4]. Laser-matter interactions in the relativistic regime have opened up various applications such as relativistic electron beam generation, fast-ion acceleration[5–14], mega-Gauss level magnetic field generation[15–17], intense X-ray[18], gamma-ray[19,20], positron[21] and neutron[22] generations, and fast-ignition-based laser fusion[23,24]. For these applications, energy absorption[25] and momentum transfer from high-intensity lasers to plasma particles are fundamental issues.

In the interaction of intense laser fields with overdense targets, the laser radiation pressure pushes electrons at the plasma surface, which sets up an electrostatic field originating from the charge separation that ultimately accelerates ions in the forward direction. The laser light proceeds to push the plasma surface, which has the relativistic critical density $\gamma n_c$ into the target. Here, $\gamma$ is the relativistic factor of electrons, $n_c = m_e \omega_L^2/(4\pi e^2)$ is the non-relativistic critical density, $\omega_L$ the laser frequency, $m_e$ the rest mass of electron, and $e$ the fundamental charge. This process is referred to as the laser hole boring (HB)[26–31]. The HB or the surface steepening has been considered to be important for applications, e.g., laser channeling[32–34], high harmonic generation[35,36], and plasma mirror[37,38]. For these applications, how long the steepened clean interface is sustained is an essential question.

Laser absorption and hot electron generation also depend on the steepening of the plasma surface in the HB process. In the conventional model of the HB[26,28–30], which is based on the momentum transfer equation from laser to ions, laser lights bore holes as long as the laser pulse continues, which implies there is no density limit for the HB.

Recently, for the laser-solid interactions in the multi-ps regime, it is reported that the plasma on the laser-irradiated front surface expands significantly in the order of 10 $\mu$m during the laser irradiation, and superthermal electrons beyond the conventional ponderomotive scaling[26] are generated[39–41]. A recent LFEX laser experiment demonstrated that the hot electron temperature increases drastically when the laser pulse duration is extended from 1 to 4 ps while keeping the peak intensity same[42]. Ion acceleration in such a multi-ps laser interaction cannot be described by the conventional target normal sheath acceleration model that assumes plasma expansion with isothermal electron temperature[43–45]. The nonisothermal plasma expansion theory is proposed to explain the multi-ps laser-driven ion acceleration experiments where the time evolution of hot electron temperature is important[46,47]. In order to control such a hot and/or superthermal electron generation in the multi-ps regime, the cause of the plasma blowout to the front side is essential to be figured out.

In this study, we find that the HB is stopped in the ps time regime even while the laser pulse is still on, and the surface plasma eventually starts to blowout to the front side. We here develop a theory that explains the transition from the HB to the plasma blowout regime. Based on a pressure balance relation between laser radiation pressure and electron thermal pressure, we derive the limit density for the HB as $8Ra_0^2 n_c \left(\sim 5.8RI_{18}\lambda_{\mu m}^2 n_c\right)$, above which the laser field cannot push beyond. Here, $a_0 = eE_0/(m_e c \omega_L)$ is the normalized laser field amplitude, $E_0$ the amplitude of the laser electric field, $R$ the reflectivity, $I_{18}$ the laser intensity normalized by $10^{18}$ W cm$^{-2}$, $\lambda_{\mu m}$ the laser wavelength normalized by 1 $\mu$m. We show the validity of the derived model by using particle-in-cell (PIC) simulations. In addition, we obtain the time scale for the transition from the HB to the blowout regime based on the momentum transfer equation in a preformed plasma[48]. The transition time

scale is found in the ps regime, and therefore, the prediction by this theory will appear in multi-ps laser experiments.

## Results

**Process of hole boring.** As an introduction of the HB, we show an interaction of plasma with linearly polarized high-intensity laser field by using Fig. 1. Here, the laser intensity is in the relativistic regime, $I = 5\times10^{18}$ W cm$^{-2}$ so that $\hat{a}_0 = 2$ where the hat indicates the peak value. We consider a thick plasma with initial ion and electron densities $n_{i0}$ and $n_{e0} = Zn_{i0}$, respectively, where $Z$ is the ion charge state. The target electron density $n_{e0}$ is an order of magnitude higher than the relativistic critical density $\gamma n_c$. We assume a pre-plasma of scale length $L$ at the front side where the electron density $n_e$ increase linearly from zero to $n_{e0}$. Figure 1a–f are the results we obtained in a PIC simulation in one-dimensional (1D) geometry. The laser field transmits through the underdense region $n_e < n_c$. After reaching the critical density $n_c$, the laser field can further penetrate into the plasma $n_e \leq \gamma n_c$ due to relativistic transparency. Since $\gamma$ fluctuates in the interaction, the plasma is not transparent completely in this regime, and thus, the laser field starts to push electrons at the pulse front. In this phase, the pile up of electrons swept up by the laser is fast enough to create a strong electric field that can accelerate ions at a speed exceeding the sound velocity. Consequently, a collisionless shock is formed at the front as seen in the ion phase plot for the longitudinal direction in Fig. 1a. Above the relativistic critical density, $n_e > \gamma n_c$, the plasma is opaque and the speed of the interaction front decreases significantly. The laser is incapable of driving shocks by pushing the overdense electrons. We refer to this stage to as the 'HB' phase. In the HB phase, electrons in the front surface are also pushed by the laser light making a charge separation. If the laser pulse can sustain the charge separation, ions start to move forward. Note that for the HB, the pulse has to be longer than the ion response time scale, $2\pi\omega_{pi}^{-1}$, which is in the order of 100 fs at the critical density. Here, $\omega_{pi}$ is the ion plasma frequency. As seen in Fig. 1b, the front of the shock generated in the relativistic transparency phase proceeds forward in a faster speed than the laser interaction front, that is the HB front.

As the HB proceeds, we find that the average ion momentum at the HB front $\overline{p}_{xf}$ changes from positive as in (b) to zero as in (c). This indicates that the HB front cannot go further when it reaches the condition that the average longitudinal density flux of ions is zero. The asymmetric relation between positive and negative ion density flow can be seen in (d) and (e) where the distributions of ion longitudinal momentum in the region of the pulse front, sampled within $\Delta x = 0.2$ $\mu$m, are shown for the corresponding times for (a) and (b), respectively. One can see that the distribution that has a notable amount of positive $p_{xf}$ component in (e) turns to a symmetrical distribution with respect to the $p_{xf}$ axis in (f). Therefore, at $t = 0.6$ ps ((c) and (f)), a stationary state of the plasma front is established, which corresponds to the stopping condition for the HB. At this time, the laser radiation pressure balances with the plasma pressure. The establishment of the stationary state is owing to the electron heating during the HB, by which a hot dense electron cloud is formed behind the HB surface, and thus the negative ion density flow is driven. Since the plasma is heated by the laser continuously, the plasma will start to expand when the electron pressure exceeds the laser radiation pressure. This corresponds to the transition from the HB to the plasma blowout. Note that the stationary state never appears in the conventional aspect of the HB where the electron heating is not taken into account.

In Fig. 2a, we show the time evolution of the positions of the critical density $n_c$ and relativistic critical density $\gamma n_c$ for the same

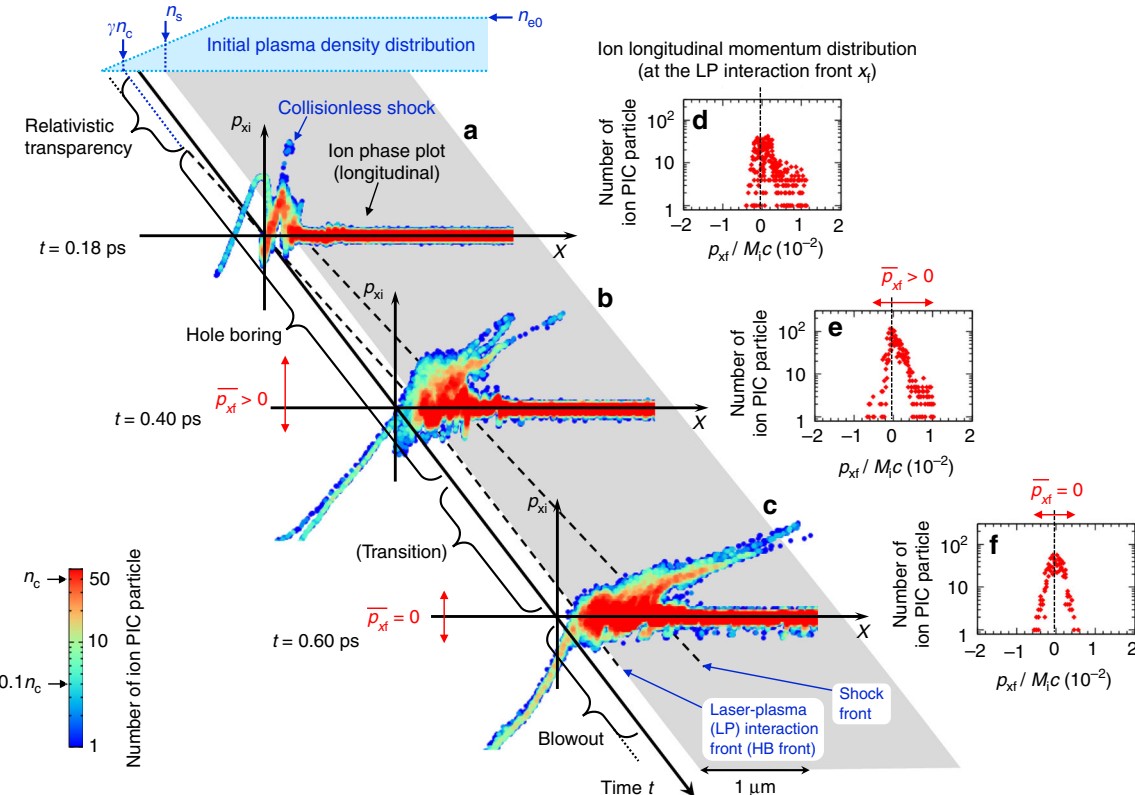

**Fig. 1** Time evolution of the interaction of solid plasma surface with a relativistic laser light. **a–f** are the results of a particle-in-cell simulation in one-dimensinal geometry. A linearly polarized laser light with the peak intensity of $5 \times 10^{18}$ W cm$^{-2}$ ($\hat{a}_0 = 2$) is irradiated from the left boundary with a 0.15 ps Gaussian pulse leading edge. A fully ionized deuteron plasma of an overdense density $n_{e0} = 40n_c$ and a 1 $\mu$m pre-plasma with linear density gradient is initially distributed. **a–c** are the phase plots of ions in the longitudinal direction at times $t = 0.18$, 0.40, and 0.60 ps, respectively, where the color represents the number of PIC particle. In **d–f**, the longitudinal momentum distributions at the laser-plasma (LP) interaction front at the corresponding times are presented. Here, the momentum $p_{xf}$ is normalized by $M_ic$ where $M_i$ is the ion rest mass

simulation of Fig. 1. The relation $\gamma \equiv \left(1 + (1 + R)\epsilon^2\hat{a}_0^2/2\right)^{1/2}$ is used where $\epsilon = 1$ for linear polarization and $\epsilon = \sqrt{2}$ for circular polarization. We here use the reflectivity $R$ at the transition time from the HB to the blowout. Figure 2b shows the reflectivity $R$, which keeps an almost constant value around the transition time from the HB phase to the blowout phase. In early time $t < 0.4$ ps, positions of $n_c$ and $\gamma n_c$ are pushed toward positive $x$ direction, however, during the time $t = 0.5$–$0.7$ ps that is shaded in Fig. 2a, b, the plasma front hardly moves from $x \sim 9.45$ $\mu$m. After this time, the plasma starts to blowout to the negative $x$ direction.

In Fig. 2d, e, we show the electron distributions in the phase space for the longitudinal $x$ direction at $t = 0.4$ ps, that is the HB phase, and $t = 1.0$ ps, that is the blowout phase, respectively. In the HB phase (d), electrons are experiencing $\mathbf{J} \times \mathbf{B}$ acceleration at the peripheral of the HB surface, $x \sim 9.4$ $\mu$m. In the low-density region in front of the HB surface, mainly the electrons in the region where $n_e < 0.1n_c$ interact directly with the incident laser field and gain momenta $p_{xe}/m_ec \sim 5$ (2.1 MeV). On the other hand, in the blowout phase (e), the region where $0.1n_c < n_e < n_c$ expands about 1 $\mu$m, which causes the enhanced acceleration of electrons[40]. The spectra of electron energy $\varepsilon_e$ at $t = 0.4$ and $1.0$ ps are shown in Fig. 2c. The number of hot electrons in the MeV range are increased by an order of magnitude, and the high-energy slope is enhanced, resulting in temperatures higher than the ponderomotive temperature $T_P$ and also than the Beg scaling $T_B$[49].

**Hole boring limit density.** We now consider the pressure balance at the laser-plasma (LP) interaction front in the stationary state, where the HB cannot proceed further. We assume that a plane

laser field with the normalized amplitude $a_0$ is irradiating an overdense plasma distributed in the region $x \geq 0$. At the stationary state, the electrons are pushed slightly by the laser radiation pressure with the distance $\ell_s/2$ where $\ell_s = \left(m_ec^2/(4\pi n_ee^2)\right)^{1/2}$ is the skin depth for the incident laser field. Note here that since the laser amplitude decays in $\ell_s$, the intensity decreases in the scale length of $\ell_s/2$. Then, only ions remain in the front surface $0 \leq x < \ell_s/2$ with the density $n_i$, while the inside $x \geq \ell_s/2$ is filled by both ions with density $n_i$ and electrons with density $n_e = Zn_i$. The positive electrostatic field generated by the charge separation at the front surface, $E_s = 2\pi en_e\ell_s$, drags ions to the positive $x$ direction. However, in the stationary state of the LP interaction front, the ion density is maintained by the negative density flow of ions as seen in Fig. 1c, f. The pressure balance relation at the interaction surface between laser field and electrons is derived by integrating the electron fluid equation of motion in the stationary state as described in the Methods as

$$(1 + R)\frac{I}{c} = n_eT_e + \frac{E_s^2}{8\pi}. \tag{1}$$

The left-hand side represents the laser radiation pressure with intensity $I$, the first term on the right-hand side (RHS) corresponds to the electron pressure with temperature $T_e$, and the second term on the RHS denotes the sheath electrostatic potential energy density, which corresponds to the surface tension in the width of the skin depth. Note that some of the previous papers have also included the electron pressure term[28,30]. However, the stationary state sustained by the surface tension

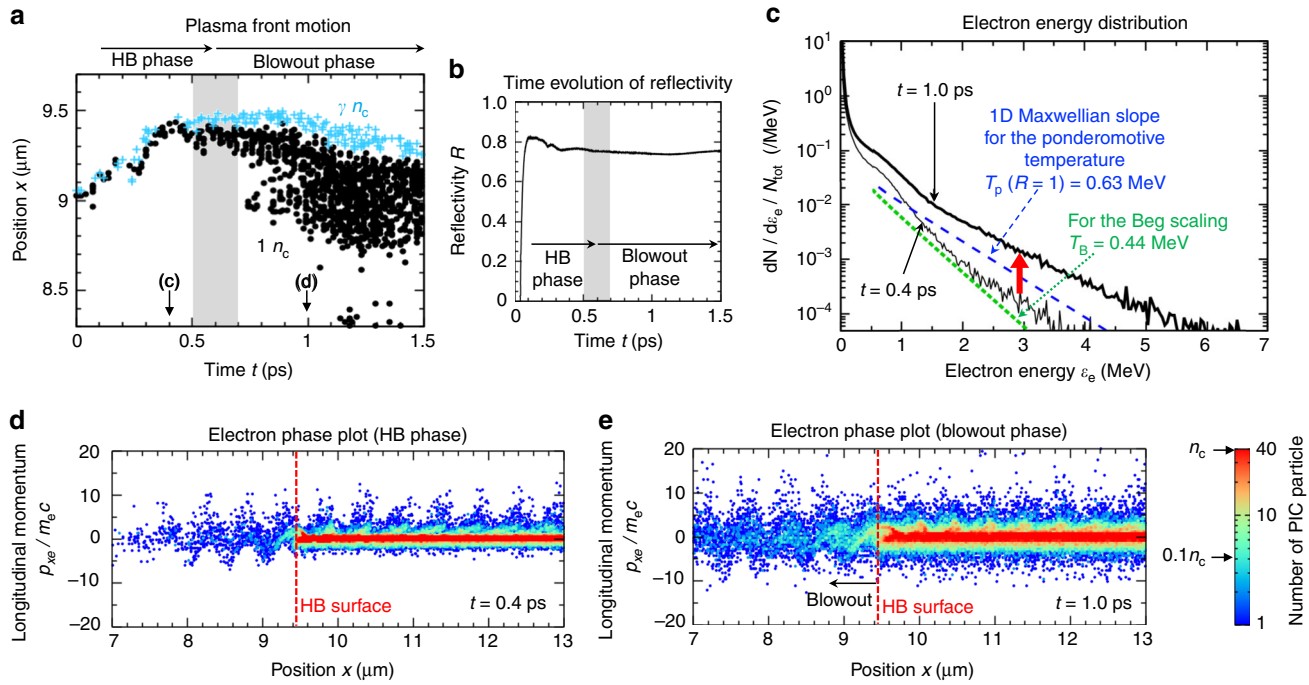

**Fig. 2** PIC simulation results for the interaction of the linearly polarized laser field. The laser intensity is the same as that in Fig. 1. **a** shows the time evolution of positions of the critical density $n_e = (1 \pm 0.1)n_c$ (black dots) and relativistic critical density $n_e = (\gamma \pm 0.1)n_c$ (blue crosses), where $\gamma = \left(1 + (1 + R)\epsilon^2 \hat{a}_0^2/2\right)^{1/2}$ is the relativistic factor of electrons and $\epsilon^2 = 1$ for linearly polarized lasers. In obtaining $\gamma$, the reflectivity $R$ at the transition time from the hole boring (HB) to the blowout is used. **b** Time evolution of the reflectivity. The shaded areas in **a**, **b** correspond to the transition interval from the HB to the blowout. In **c**, the electron energy distributions at $t = 0.4$ and 1 ps are shown by black thin and bold lines, respectively. The slopes of the one-dimensional relativistic Maxwellian energy distribution function >0.5 MeV are shown by the blue-dashed and green-dotted lines assuming the ponderomotive temperature at the maximum reflectivity $R = 1$, that is $T_p = 0.63$ MeV, and the Beg scaling $T_B = 0.44$ MeV, respectively. **d**, **e** are the electron phase plots in the longitudinal $x$ direction at times $t = 0.4$ and 1 ps, respectively

of the field has never been considered, which is the critical difference of Eq. (1) from the conventional descriptions of the HB. We assume that the plasma is composed of hot and bulk electron components as $n_e = n_h + n_b$. We here assume $n_e T_e \sim n_h T_h$ where $T_h$ is the hot electron temperature, by neglecting the bulk electron pressure.

The hot electron density and velocity can be determined by the conservation of energy density flux,

$$(1 - R)I = \alpha n_e T_e c \beta_e. \tag{2}$$

Here, $\beta_e = v_e/c$ is the ratio between electron drift velocity $v_e$ and the speed of light $c$, and $\alpha \equiv ir/2$ is the geometrical factor where $r = 1$ for the non-relativistic Maxwell momentum disrtibution, $r = 2$ for the relativistic Maxwell (Maxwell–Jüttner) momentum distribution, and $i = 1, 2,$ or 3 represents the dimension of momentum distribution. Hereafter, we consider $r = 2$. For quasi-1D relativistic interactions, we can assume $i = 1$, and thus, $\alpha = 1$. Equation (2) indicates that the absorbed laser energy flux is carried by electrons. We simplify Eq. (2) by replacing $n_e T_e \beta_e$ on the RHS by $n_h T_h$. This approximation is valid for the relativistic condition, $a_0 > 1$ and thus $\beta_h \sim 1$. When the laser field amplitude becomes super relativistic ($a_0 \gtrsim 300$), other energy loss mechanisms such as the radiation damping will come into play, and hence, the pressure balance at the LP interaction surface will change[50].

From Eqs. (1) and (2), we can derive the front density $n_e$ for the stationary state. We define this stationary density as the limit density of the HB, $n_s$, which is obtained as

$$\frac{n_s}{n_c} = 8\epsilon^2 a_0^2 \frac{1 + R - (1 - R)\beta_h^{-1}\alpha^{-1}}{2}. \tag{3}$$

Here, we used the relation $I/c = n_c m_e c^2 \epsilon^2 a_0^2/2$. Equation (3) represents the limit density for the HB, above which the laser light cannot proceed forward. For this limit density, the laser light with intensity $I$ is incapable of sustaining a charge separation that is sufficient for driving positive mean density flux of ions. When one assumes a 1D relativistic Maxwell distribution $\alpha = 1$, the relativistic limit for the hot electron velocity $\beta_h = 1$, and linear polarization $\epsilon = 1$, Eq. (3) reduces to

$$\frac{n_s}{n_c} = 8Ra_0^2. \tag{4}$$

Equation (4) presents the dependence of the HB limit density on normalized laser amplitude $a_0$ and reflectivity $R$. Note that for the ideal condition for the HB, we generally consider the density regime $n_e \sim \gamma n_c \sim a_0 n_c$. On the other hand, the HB limit density $n_s$ given by Eq. (4) corresponds to a higher density than $\gamma n_c$ for $a_0 > 1$ when the reflectivity is not so low satisfying $R > a_0^{-2}/8$. In other words, the HB proceeds beyond the density regime $n_e \sim a_0 n_c$, and stops at the density $n_e = 8Ra_0^2 n_c$. The maximum HB limit density is obtained for $R = 1$, that is $n_{smax}/n_c = 8a_0^2$. In the case of multi-dimensional interactions ($\alpha > 1$), due to the additional freedom of lateral energy diffusion, the plasma pressure becomes lower even with the same $R$ as seen from Eq. (2). Consequently, laser lights can proceed higher density in multi-dimensional geometry than in the 1D case. However, the maximum limit density $8a_0^2 n_c$ is not changed by the dimension effect as can be confirmed by Eq. (3).

In Fig. 3, we plot Eq. (4) for various reflectivities $R$ by dotted lines. The bold solid line for $R = 1$ corresponds to the maximum HB density. Hence, the shaded area above the solid line is the

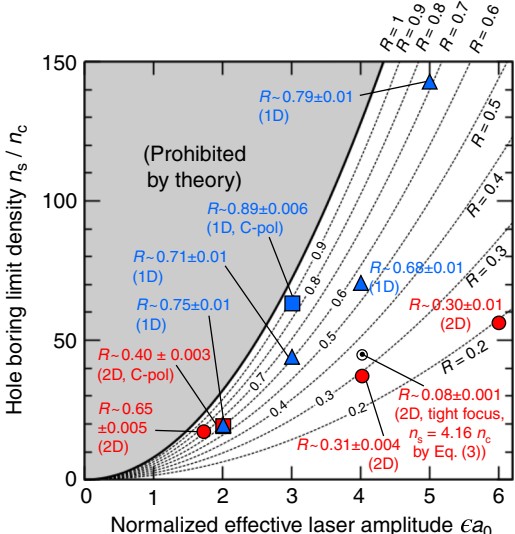

**Fig. 3** Dependence of the hole boring limit density $n_s$ on the normalized effective laser field amplitude $\epsilon a_0$. Here, $\epsilon = 1$ and $\sqrt{2}$ for linear and circular polarizations, respectively. Theoretical lines given by Eq. (4), that is for the 1D relativistic condition $\alpha = 1$ and $\beta_h = 1$, are presented by black solid and dotted lines for various reflectivities $R$. The shaded area corresponds to that prohibited by the theory for any reflectivity $R$. Blue triangles and blue square show the PIC simulation results for one-dimensional geometry with linear and circular polarizations, respectively. Red circles and red square are the 2D PIC simulation results for the quasi-1D geometry with widely focused (spot size of 60 μm) linear-P and circular polarizations, respectively. The white circle corresponds to the PIC simulation result for a tightly focused (spot size of 1.5 μm) P-polarized laser pulse in two-dimensional geometry, for which the corresponding theoretical value of $n_s$ assuming the 2D relativistic condition $\alpha = 2$ and $\beta_h = 1$ in Eq. (3) is presented. Reflectivity $R$ obtained in each simulation is also shown

prohibited area where laser light cannot reach that density. We also present the maximum density to which the laser light can reach in 1D and two-dimensional (2D) (quasi-1D) PIC simulations. We see that all the simulation points are below the maximum HB limit density. The maximum density obtained in each simulation agrees with the theoretically obtained limit density $n_s$ for $R$ with a difference limited in the range of $\pm 0.1$, except for the result of 2D circular polarization (C-pol), which will be addressed below. Note that in the 2D geometry, the reflectivity tends to be lower than the 1D geometry, so that the maximum density in the 2D simulation is typically far below the theoretical limit, that is $n_{smax}$ for $R = 1$. Although the reflectivity $R$ changes in time, the variation of $R$ in the shaded interval in Fig. 2b, that is $\pm 0.1$ ps, around the HB stationary state is small about 0.01 as shown in Fig. 3 for each simulation.

Hereafter, we estimate the time scale $t_s$ to reach the HB limit density. As seen in Fig. 2, superthermal electrons appear after the transition to the blowout phase. Hence, $t_s$ is important to control the electron heating, which is essential for various applications of intense LP interactions.

During the HB stage, the momentum transfer from laser light to ions, in the frame moving with the ion front velocity $v_f$, is expressed by

$$(1 + R)\frac{I}{c} = 2n_i M_i v_f^2, \tag{5}$$

where $M_i$ is the ion mass, and the electron pressure term is neglected for simplicity. Here we assume an exponential density profile with the scale length $L$ as $n_e = Zn_i = \gamma n_c \exp(x/L)$. The front

position of the HB ($x_f$) is obtained by integrating $v_f$ in time as

$$x_f = 2L \ln\left[1 + \frac{ct}{2L}\left(\frac{m_e Z}{2M_i}\frac{\gamma^2 - 1}{\gamma}\right)^{1/2}\right], \tag{6}$$

as in ref. [48]. Then, the electron density at the HB front $n_f$ is given as

$$n_f = \gamma n_c \exp(x_f/L). \tag{7}$$

The transition time scale $t_s$ is obtained by solving Eqs. (6) and (7) for $t$ with substituting $n_s$ to $n_f$ in Eq. (7) as

$$t_s = \frac{1}{\sqrt{\frac{1+R}{2}}\epsilon a_0}\frac{L}{Ac}\left(\sqrt{\frac{n_s}{n_c}} - \sqrt{\gamma}\right), \tag{8}$$

where $A \equiv (Zm_e/(2M_i))^{1/2}/2$, and $n_s/n_c$ on the RHS is given by Eq. (3). Here, we define the time when the laser front reaches to the position of $n_e = \gamma n_c$ as $t = 0$. In the above derivation, we assume the laser field amplitude $a_0$ is constant in time. Note that this assumption is valid for the present simulations where the pulse profile is constant in most of the interaction time. In the case of laser pulses with Gaussian temporal profiles, the time scale for reaching the HB limit density $n_s$ is estimated approximately to be $2t_s$. Equation (8) implies that the LP interaction front proceeds by boring the exponentially distributed plasma with the initial scale length $L$, and then subsequently reaches the HB limit density $n_s$ in the time scale of $t_s$.

In Fig. 4a, the transition times $t_s$ given by Eq. (8) are plotted as a function of $\epsilon a_0$ by black lines for various reflectivities $R$. Here, the relativistic 1D momentum condition, that is $\alpha = 1$ and $\beta_h = 1$, and pre-plasma scale length $L = 2$ μm are used. The derived time scales are in the order of ps. The times when the HB stops in the simulations agree well with the theoretical lines. In the case of circular polarization, which is the combination of P- and S-polarizations, the P-polarization has higher absorption than the S-polarization, resulting that the polarization of the laser field during the interaction is not circular anymore. Namely, the interaction becomes a two-beam-like interaction, so that the result departs from the single beam scaling, e.g., $n_s$ and $t_s$.

As an example of the HB process, Fig. 4b shows the electron density profile in position-time space obtained by the 2D simulation shown in Fig. 4a at $\epsilon a_0 = 4$. The laser front gets to the position of $n_e = \gamma n_c$ at $t = 0$ by the process of relativistic transparency. Up to time $t = t_s$, the HB surface moves to the higher density direction. After passing $t = t_s$, the HB surface is pushed backward (lower density side), which corresponds to the plasma blowout.

To check the multi-dimensional effect, we demonstrated a 2D simulation for a tightly focused laser pulse. The results are plotted in Figs. 3 and 4 by white circles. Both results are in a good agreement with values derived by Eqs. (3) and (8), assuming the 2D condition $\alpha = 2$ and reflectivity $R$ obtained in each simulation. The obtained scalings are therefore confirmed to be applicable in multi-dimensional situations.

In conclusion, we derived the limit density for the HB $n_s$ based on a balance relation between laser radiation pressure and plasma pressure. The HB limit density is found to be $n_s = 8Ra_0^2 n_c (\sim 5.8RI_{18}\lambda_{\mu m}^2 n_c)$. After reaching the HB limit density $n_s$, the laser light is incapable of sustaining the charge separation that is sufficient for driving the forward mean density flux of ions in the HB surface, and thus, the laser light can no longer proceed into the higher density region. By using the PIC simulation, we demonstrated that the HB reaches the stationary state when the laser pulse front proceeds to the density equal to

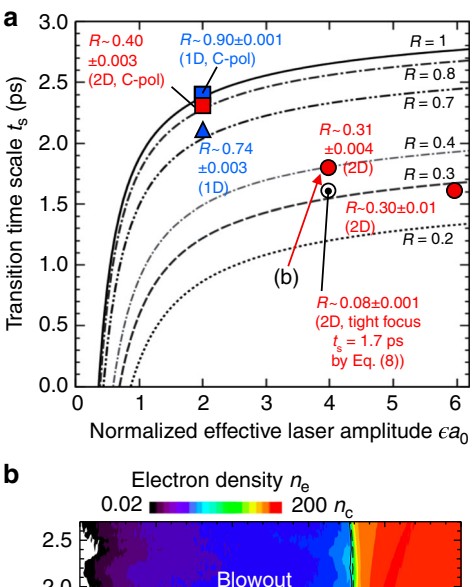

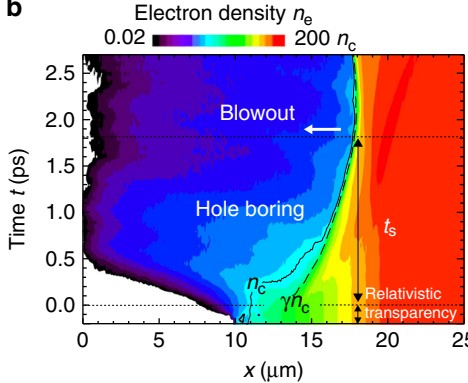

**Fig. 4** Scaling of the transition time $t_s$ on the laser amplitude. **a** Dependence of the transition time from the hole boring to the blowout, $t_s$, on the normalized effective laser field amplitude $\epsilon a_0$ for various reflectivities $R$ where $\epsilon = 1$ and $\sqrt{2}$ for linear and circular polarizations, respectively. Here, the relativistic 1D condition, that is $\alpha = 1$ and $\beta_h = 1$, and pre-plasma scale length $L = 2\,\mu m$ are used. Theoretical lines given by Eq. (8) are presented by black lines. Red circles and red square are the 2D PIC simulation results for the quasi-1D geometry with widely focused (spot size of 60 μm) linear-P and circular polarizations, respectively. The blue triangle (square) is the PIC simulation result for one-dimensional geometry with linear (circular) polarization. The white circle corresponds to the PIC simulation result for a tightly focused (spot size of 1.5 μm) P-polarized laser pulse in two-dimensional geometry, for which the corresponding theoretical value of $t_s$ assuming the 2D relativistic condition $\alpha = 2$ and $\beta_h = 1$ in Eq. (8) is presented. Reflectivity $R$ obtained in each simulation is also shown. **b** Electron density profile in position-time space obtained by the 2D simulation shown in **a** at $\epsilon a_0 = 4$ with the wide laser focal spot. The presented density is the averaged value over the central region $y = \pm 1\,\mu m$ with respect to the laser axis. Black solid and dashed curves are contour lines for critical density $n_c$ and relativistic critical density $\gamma n_c$, respectively

the limit density $n_s$. We derived the transition time $t_s$, at which the plasma turns from the HB to the blowout, as Eq. (8).

The transition time $t_s$ is critical for applications such as laser channeling, high harmonic generation, and plasma mirror, which require steepened clean interface for efficient operation. The blowout plasma interacts with the laser field resulting copious superthermal electrons, which are important to increase the efficiency of the applications, e.g., ion acceleration[46,47] and pair-plasma creation[51,52] with multi-ps kiloJoule laser lights.

## Methods
**Numerical simulations**. Simulation results in 1D geometry shown in Figs. 1,2–4 are obtained using a fully relativistic collisional PIC code EPIC3D[53]. The calculations are executed in 2D spatial dimension where the size of simulation box is $L_x$

$= 40.96\,\mu m$ in the laser propagation direction, while we use four meshes in the transverse $y$ direction with the mesh size of 10 nm. A laser field with the normalized amplitude of $a_0(t) = \hat{a}_0 f_a(t)$ is excited by an antenna at the left boundary. Here, $\hat{a}_0$ is the peak value and $f_a$ is the pulse shape factor whose time dependence at the antenna position is given by $f_a = \exp\left[(t - t_0)^2/\tau_L^2\right]$ for $t < t_0$ and $f_a = 1$ for $t \geq t_0$, where $t_0 = 0.1$ ps and $\tau_L = 0.15$ ps. The laser wavelength is $\lambda_L = 1.05\,\mu m$. In the calculations for Figs. 1 and 2, the peak normalized amplitude is $\hat{a}_0 = 2$, which corresponds to the intensity of $5 \times 10^{18}\,W\,cm^{-2}$. A uniform fully ionized neutral deuteron plasma is distributed initially from $x = 10–20\,\mu m$ with a linear pre-plasma of length $L = 1\,\mu m$ whose electron density increases from zero at $x = 9\,\mu m$ to the uniform plasma density $n_{e0} > 8\hat{a}_0^2$ at $x = 10\,\mu m$. In the calculations for Figs. 1 and 2, $n_{e0} = 40n_c$ is assumed. The initial plasma distribution and the incident laser field are uniform in the transverse $y$ direction.

Simulation results in 2D geometry shown in Figs. 3 and 4 are obtained using a fully relativistic PIC code PICLS[54]. The size of simulation box is $L_x = 40\,\mu m$ in the laser propagation direction and $L_y = 120\,\mu m$ in the transverse direction with the mesh size of 20 nm. The laser pulse function and the wavelength are same as those used in the EPIC simulations. A uniform fully ionized neutral deuteron plasma is distributed initially from $x = 20–40\,\mu m$ with an exponential pre-plasma of scale length $L = 2\,\mu m$ whose electron density increases from $0.6n_c$ at $x = 10\,\mu m$ to the uniform plasma density $100n_c$ at $x = 20\,\mu m$. The initial plasma distribution is uniform in the transverse $y$ direction. The laser field is focused at $x = 20\,\mu m$ to the spot sizes of 60 μm in the wide focus case, and 1.5 μm in the tight focus case, which is close to the diffraction limit to reduce the self-focusing effect. Since the spot size for the wide focus cases is much larger than the plasma scale length $L$, the interaction around the beam center can be regarded as quasi-1D, that is $\alpha \sim 1$, where the effect of multi-dimensional energy diffusion is small.

The reflectivities $R$ for the simulations referred in Figs. 3 and 4 are observed around the transition time $t_s \pm \Delta t$, where $\Delta t = 0.1$ ps, which corresponds to the scale of the ion response time $2\pi\omega_{pi}^{-1}$.

**Derivation of the pressure balance equation**. The pressure balance relation for the stationary state of the HB Eq. (1) is derived from the electron fluid equation of motion in the stationary state given by

$$0 = -\frac{m_e c^2}{4\gamma} n_e \epsilon^2 (1 + R) \nabla a_0^2 - \nabla(n_e T_e) + e n_e \nabla\phi, \quad (9)$$

where $\phi$ is the electrostatic potential, and all quantities are averaged over the laser period. Here, we consider a 1D geometry where an overdense plasma is distributed in the region $x \geq 0$. Electrons are pushed by the laser radiation pressure with the distance $\ell_s/2$, where $\ell_s$ is the skin depth, while ions remain in the front surface $0 \leq x < \ell_s/2$ with the density $n_i (= n_e/Z)$. We assume the scales of spatial variation $\nabla n_e \sim 2n_e/\ell_s$ and $\nabla a_0 \sim -a_0/\ell_s$ in the interface, and obtain $\nabla(n_e a_0^2/\gamma) = -n_e(\nabla a_0^2)/(2\gamma)$. Here, $\gamma \equiv \left(1 + (1 + R)\epsilon^2 \hat{a}_0^2/2\right)^{1/2}$ is used, and we approximated $(1 + R)\epsilon^2 \hat{a}_0^2/2 \sim \gamma^2$ in the calculation of $\nabla\gamma$. Using the above assumptions for spatial variation, integrate Eq. (9) from $x = -\infty$ to $x = +\infty$, then

$$(1 + R)\frac{I}{c} = n_e T_e + \frac{E_s^2}{8\pi}, \quad (10)$$

which is the pressure balance relation Eq. (1). Here, the relation $I/c = n_c m_e c^2 \epsilon^2 a_0^2/2$ is used, and the electron density at the laser front is given by $\gamma n_c$. $E_s^2/8\pi$ is the sheath electric field energy density generated by the charge separation in the interface $0 \leq x < \ell_s/2$. In obtaining this sheath field term, we used the Poisson equation and neglected the ion pressure.

Note that in the HB stage where the ion front has not reached the stationary state, the charge separation field eventually moves the ion front in the forward (positive $x$) direction. Therefore, for describing the HB stage, the last term on the RHS of Eq. (10) is replaced by the kinetic energy density of ions moving with the HB velocity $v_f$, and then the momentum transfer equation in the frame moving with the velocity $v_f$ is obtained as Eq. (5).

**Data availability**. The data that support the findings of this study are available from the corresponding author upon reasonable request.

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

## Acknowledgements

We are grateful for Dr. S. C. Wilks for fruitful discussions and encouragement. This study was supported by JSPS KAKENHI grant numbers JP15K21767 and JP15K17798, and also by NIFS Collaboration Research program (NIFS17KNSS090).

## Author contributions

N. I. and Y. S. developed the analytical theory with support from K. M. Assistance with conceptualizing the physical arguments was provided by S. K. Numerical simulations were performed by N. I.,Y. S. and M. H.

## Additional information

**Competing interests:** The authors declare no competing financial interests.

