## [Peer Review File · Nature Communications]

Reviewers' comments:

Reviewer #1 (Remarks to the Author):

These results will be of interest to people in the larger community because the heating of electrons and the amount of absorption are fundamental to all intense laser-plasma interactions, and many applications that employ these lasers (i.e., plasma spectroscopy, particle generation, novel fusion ideas, plasma-based light sources, ...) all care about these things.

Specifically, Figure 1 is exceptionally informative and clear. The authors are depicting a difficult, highly nonlinear and rapidly changing series of stages in this complex phenomenon, and this figure captures the essence of the various stages and presents the progression in a clear way with clean graphics. Indeed, the entire article can be summed up in this figure, and immediately the reader can appreciate the evolution from hole-boring to the blowout regime, and the consequences this brings to understanding both the absorption and particle energies that result during the interaction. The authors then go on to derive the time over which this takes place, and the density up to which hole boring can occur.

There is an issue with Eq (1). The words above imply that they are showing momentum conservation in the equation, but the equation shown looks like the light pressure being balanced by the pressure in the plasma, but perhaps I don't understand the English so well here. Typically, this equation results from balancing the momentum flow with the light pressure. I would request the authors reword the sentence prior to equation 1 to make it clear to the reader exactly what equality they are writing down here, and produce more detail as to where each term comes from. The equation is correct, I just think what they are calling it is confusing in the text.

Figure 3 is problematic from an uncertainty point of view. The simulation points on the plot, while in rough agreement with the analytic model, do have substantial variations to the actual calculated values. While the overall trends are always correct, specific values often vary by a considerable amount, yet others agree perfectly (even for the 1D; the authors acknowledge and discuss why the 2D simulations do not agree.) It would have been helpful if error bars for the simulations were included. This is difficult, but a rough idea of what they are would be extremely useful. I suspect that the reflectivity was not one single number in the simulations, (and correspondingly, the HB limit density may "bounce" around a bit, and not be a single value, as currently is shown with a single point) and therefore a range could potentially be obtained from this. The same could be said for Figure 4: is there no error on the actual time? I am surprised by this, and would have thought that this time was approximate, and not a precise quantity. Looking at Fig 4(b) it certainly looks like there is an entire region where one could have obtained a t_s . Perhaps this is the uncertainty? In any event, this is for the authors to determine. In my opinion, this uncertainty issue is the main problem I have with the entire article and should be addressed by the authors. Finally, there is no 1D result at a normalized laser amplitude of 6. Is this because it did not agree with the theory? If not, this is good to know.

This last point is related to the second most important shortcoming of the paper, which is that the bounds of applicability are not clearly explained. Throughout the paper, certain approximations are made, and there is some discussion of the density regime in which HB is ideal, but how high can the laser amplitude go before the results break down? There is some discussion of this after Eq. (4), and perhaps just making this argument clear will help. For example, I do not understand this sentence, due to grammar: "Consequently, laser lights proceed higher density than the 1D case at the same reflectivity of $R < 1$." Throughout the paper, assumptions are made, but it is not clear that the authors sufficiently state under what conditions this is satisfied for their results to hold. It is certainly satisfied

in the regime that they do their simulations that are presented. But the region of applicability needs to be explicitly stated. Experimentalists will look at the results here, plug in their values of 400 n_{cr} , a 2D spot, and wonder why they get crazy results. Therefore, I recommend that the limits of applicability be more explicitly stated.

It would also have been nice to see the actual reflectivity for the simulations. It appears, from the discussion and figures in the article, that it is a simple number, and that is the end of the story. However, I can easily imagine the reflectivity varying wildly over the time of incidence. Does this contribute to the uncertainty in the values of reflectivity that we see plotted on the figures?

In summary, the conclusions about the density limit and the onset of a blowout regime at the end of hole boring in the manuscript are original, in that although things like hole boring have been investigated in the past, none have taken them to their logical conclusion. In other words, when the simple formulae for hole boring break down, what happens at the laser-plasma interface in the long term has not previously been addressed in as much detail as this paper does. This paper puts forth not only simulations that show what happens, but a predictive formula (for a set of laser-plasma parameters) that describes the long-time behavior. The paper will certainly influence the thinking on this specific "saturation" mechanism of hole boring and what densities are achievable (for these particular parameters) and should stimulate thinking about what happens when more complexity is introduced into the problem, like 3-D motion, high density targets where the bulk electron pressure cannot be neglected, radiation losses, and so on. As discussed above, one weakness is that it appears that each simulation has an exact value of reflectivity, and therefore the actual fluctuations in the PIC simulation values is not quantified. However, the trends certainly agree with the trends predicted by the analytic theory. It is satisfying that very specific predictions are made and that those predictions are testable by both experiment and simulation.

Reviewer #2 (Remarks to the Author):

Referee's report on NCOMMS-17-20955-T:
Plasma density limits for hole boring by intense laser pulses
by N. Iwata et al.

Hole boring by the laser beam pressure has become a fashionable scheme for inertial fusion in connection with cone guided fast ignition at comparatively moderate laser intensities (see M. Tabak). For low coronal densities, far too low for fusion applications, hole boring has been shown to work (A. Pukhov and J. Meyer-ter-Vehn). Later, about one and half decade ago, numerical studies have shown that at laser intensities below 10^{20} W/cm² hole boring was not a realistic scheme because it stopped at low density of the compressed pellet; Tabak's predictions were too optimistic. However, again a decade later new studies at some 10^{22} W/cm² made fusion with hole boring again possible, not to say fashionable, with sub-ps pulses (N. Naumova, T. Schlegel et al., 2009).

On such a background the present paper has to be considered when its authors claim that hole boring with ps pulse length comes to a rest at a limiting plasma density. Their claim is founded on analytics and on 1D and 2D PIC simulations. I have no difficulty to verify the fundamental relations (1) – (3) from which the assertion follows, except the assumption $I/c = \dots$ after eq. (3) may be a bit optimistic, although it does not appear unreasonable. How sensitive is the result on this assumption? There is another point which in the referee's opinion needs real clarification. Have a look at Fig. 2(a) and (b) of N. Naumova, T. Schlegel et al., 2009, with piston velocities for $a_{circular}$ from 20 to 300:

No stopping or recession is discovered! This seems to be in contradiction to the claim of the present paper. This point should be clarified.

A final remark. It has become a custom to reference "Wilks scaling" in connection with energetic electron generation. There exist numerous other scaling laws, all of them of doubtful value. However, among them, Wilks' scaling is the worst.

CONCLUSION: The paper submitted by N. Iwata et al. contains a very important statement. It will have a major impact on forthcoming studies on inertial fusion. It is presented in a clear way, the reader can follow all steps. No doubt, after clarification of the two points above the paper deserves publication in NCOMMS.

Reviewer #3 (Remarks to the Author):

The manuscript by Iwata et al. entitled "Plasma density limits for hole boring by intense laser pulses" describes a simple and elegant theoretical estimate of the maximum electron density possibly reached by laser hole boring, depending on the laser intensity and the plasma reflectivity. The authors back up their approach with PIC simulation results, approximately agreeing with their theoretical predictions.

Except from a few typos and unclear sentences, the manuscript quality is good, and their approximations seem justifiable (although not always explicitly justified in the manuscript).

I believe that the current manuscript does not provide sufficient insight into what has substantially changed from previous theories. It appears that no other authors had previously attempted to estimate this maximum density, even though there have been more complete and detailed theories of hole boring. It is difficult to judge the article's novelty without a better comparison. A hint of explanation seems to be given at line 16, saying that previous approaches were "based on the momentum transfer equation from laser to ions", causing no density limit. Later, at lines 31 and 85, they claim to use a pressure balance equation, and invoke the electron fluid equation of motion at line 95, that they unfortunately did not detail nor reference. What has changed compared to the previous approaches? My understanding is that the novelty lies in the electron pressure term that has been introduced, revealing that the momentum flux eventually leaks into hot electrons, preventing hole boring to continue. This aspect has not been made clear.

A second, more important point, that the authors have not made, is why their elegant formula would impact any of the applications they have listed. The participation of hot electrons in the momentum conservation is certainly an interesting point to make, but its impact is not obvious. I believe that, for publication in Nature Communications, the authors should put their result in the perspective of its outcomes in related studies. Some justification attempts are given in the introduction. For instance, line 6, "energy absorption and momentum transfer ... are fundamental issues". But how did this article address this problem? Another example, line 26, "the plasma blowout to the front side is essential to be figured out". Again, I don't see how this article has brought new insight here. None of the cited applications (channeling, high harmonics, plasma mirrors) are directly examined or even commented in the light of the authors findings. In its current form, I cannot see any "critical" advance (requested by Nature Communications) in the understanding of the hole-boring theory or in its applications. Among the proposed applications, I only see fast-ignition benefiting from this result, as the maximum density plays an important role in its design; the authors also provide a way to decide the laser optimal duration, which is handy. However, the formula only attempts at providing estimates, made even more uncertain due to the unknown laser reflectivity. This hardly provides a critical advance in

fast-ignition design.

To conclude, I find this paper of good quality and leads to elegant and handy formulae. It brings interesting insight in the momentum balance for hole boring (although this aspect should be discussed more). My main objection is that the authors have not shown why this has sufficient impact in the community. Without an explicit critical advance, which I understand is necessary for publication in Nature Communications, I would recommend for publication in a more specific journal.

In addition to the major and general comments above, I list here a few particular remarks.

1/ line 10: To my knowledge, most people in the field employ the term "critical density" instead of "cutoff density". In other fields, one can encounter "cutoff frequency", but this seems rare in this community. I suggest a note be given to ensure universal understanding.

2/ line 32: The equal sign in the parentheses should be an approximately equal sign. This applies also line 174.

3/ line 71: How is R chosen? The manuscript states "We use the reflectivity R at the transition time from the HB to the blowout". Does that mean R is considered constant and the value is somehow extracted from the simulation at a time between 0.4 and 0.6 ps? And, instead of this calculation of the relativistic critical density, why not simply measure the place where there is no more laser field?

4/ line 79: The expression "electrons with densities $n_e < 0.1 n_c$ " should be corrected. The authors certainly refer to "the electrons in the region where $n_e < 0.1 n_c$ ". This appear again line 80.

5/ line 92: The expression of the charge separation field E_s appears wrong. I believe it is $E_s = 4 \pi e n_e l_s/2$. However, the subsequent calculations are correct.

6/ line 95: The quantity ϕ should be defined.

7/ line 101: The quantity T_c should be T_e .

8/ Figure 3: Having 2D simulations compared to 1D predictions is troubling. The authors could consider duplicating their figure for clarity: one panel for 1D and one panel for 2D.

9/ line 137: "Hereafter, we estimate the time scale t_s to reach the HB limit density. This time t_s corresponds to the time scale of the HB after that the plasma starts to blowout." The two cited sentences seem to contradict each other. The first sentence seems to define t_s as the time between the beginning of the interaction up to the HB density limit. The second sentence seems to define t_s as the time between the HB limit, and the appearance of a blowout plasma. I believe the second sentence is ambiguously constructed.

10/ Fig 4a: Why is this figure showing less points than Fig 3? Did the authors verify their theory only for some of their simulations? Or is there a problem with the other simulations that are not present in Fig 4a?

Reply to Reviewer #1

Manuscript Number: NCOMMS-17-20955-T

Title: Plasma density limits for hole boring by intense laser pulses

Authors: N. Iwata, S. Kojima, Y. Sentoku, M. Hata and K. Mima

Dear Reviewer #1:

We wish to express our appreciation to the reviewer for valuable comments and insightful suggestions, which have helped us improve the paper. The following is a point-by-point response to the reviewer's comments.

Note that the reviewer's comments are shown in the box, sentences from the manuscript are written in *italic* and enclosed in double quotation marks “ ”. The revised parts are colored in **red**.

Comment 1:

‘These results will be of interest to people in the larger community because the heating of electrons and the amount of absorption are fundamental to all intense laser-plasma interactions, and many applications that employ these lasers (i.e., plasma spectroscopy, particle generation, novel fusion ideas, plasma-based light sources, ...) all care about these things.

Specifically, Figure 1 is exceptionally informative and clear. The authors are depicting a difficult, highly nonlinear and rapidly changing series of stages in this complex phenomenon, and this figure captures the essence of the various stages and presents the progression in a clear way with clean graphics. Indeed, the entire article can be summed up in this figure, and immediately the reader can appreciate the evolution from hole-boring to the blowout regime, and the consequences this brings to understanding both the absorption and particle energies that result during the interaction. The authors then go on to derive the time over which this takes place, and the density up to which hole boring can occur.’

We greatly appreciate for evaluating our work as a fundamental study that will attract attentions from people in the larger community in laser plasma physics and applications. We also thank you for the positive comments on Fig. 1 in which we intend to show what we are considering in the whole interaction.

Comment 2:

‘There is an issue with Eq (1). The words above imply that they are showing momentum conservation in the equation, but the equation shown looks like the light pressure being balanced by the pressure in the plasma, but perhaps I don't understand the English so well here. Typically, this equation results from balancing the momentum flow with the light pressure. I would request the authors reword the sentence prior to equation 1 to make it clear to the reader exactly what equality they are writing down here, and produce more detail as to where each term comes from. The equation is correct, I just think what they are calling it is confusing in the text.’

We agree with the reviewer's opinion that Eq. (1) is to be referred to as the pressure balance equation. The reason we had called it “momentum conservation” is that we derived Eq. (1) from the

electron momentum equation that is averaged over the laser period:

$$n_e \frac{dp}{dt} = -\frac{m_e c^2}{4\gamma} n_e \epsilon^2 (1 + R) \nabla a_0^2 - \nabla (n_e T_e) + e n_e \nabla \phi$$

Because we here assume the stationary state, $dp / dt = 0$, the remaining terms indicate the pressure balance as the reviewer mentioned. We have revised the sentence prior to Eq. (1) from

“*The momentum conservation at the interaction surface between laser field and electrons is*”

to

“*The pressure balance relation at the interaction surface between laser field and electrons is*”
(line 31 on page 4).

In addition, we have added the derivation of Eq. (1) in the **Methods** section as “**Derivation of the pressure balance equation**”.

Comment 3:

‘Figure 3 is problematic from an uncertainty point of view. The simulation points on the plot, while in rough agreement with the analytic model, do have substantial variations to the actual calculated values. While the overall trends are always correct, specific values often vary by a considerable amount, yet others agree perfectly (even for the 1D; the authors acknowledge and discuss why the 2D simulations do not agree.) It would have been helpful if error bars for the simulations were included. This is difficult, but a rough idea of what they are would be extremely useful. I suspect that the reflectivity was not one single number in the simulations, (and correspondingly, the HB limit density may “bounce” around a bit, and not be a single value, as currently is shown with a single point) and therefore a range could potentially be obtained from this. The same could be said for Figure 4: is there no error on the actual time? I am surprised by this, and would have thought that this time was approximate, and not a precise quantity. Looking at Fig 4(b) it certainly looks like there is an entire region where one could have obtained a t_s . Perhaps this is the uncertainty? In any event, this is for the authors to determine. In my opinion, this uncertainty issue is the main problem I have with the entire article and should be addressed by the authors.’

We appreciate the insightful comment. As the reviewer pointed out, the reflectivity R changes during the interaction. In Fig. A on the next page, we show two examples of the time evolution of the reflectivity R in the 1D PIC simulations. We can see that except for the early time of the HB, the reflectivity R takes an almost constant value. Here, we checked the reflectivity at times $t_s - \Delta t$, t_s and $t_s + \Delta t$ where $\Delta t = 0.1$ ps. This time width Δt corresponds to the scale of the ion response time $2\pi\omega_{pi}^{-1}$, as is noted in the manuscript as “.... the ion response time scale, $2\pi\omega_{pi}^{-1}$ which is in the order of 100 fs at the critical density.” (lines 47-49 on page 2). The time variation of the reflectivity R within the time Δt was 0.01 in both (a) $a_0 = 2$ and (b) $a_0 = 5$ cases. This variation of R could correspond to the uncertainty the reviewer mentioned.

According to the above analysis, we have added the variation of R within $\Delta t = 0.1$ ps for each simulation points on Figs. 3 and 4. We have found that the variation of R during the time range $t_s \pm \Delta t$ ($\Delta t = 0.1$ ps) is 0.01 or smaller for all of the simulations shown in Fig. 3. We have also added a sentence to explain how (for which time) we obtained R in the simulations in the **Methods** section as

Fig. A: Time evolution of reflectivity R in the 1D PIC simulations for the cases of (a) $a_0 = 2$ and (b) $a_0 = 5$. Around the transition time from the HB to the blowout, t_s , the time variation of the reflectivity is small, i.e., 0.01 in the time range of $t_s \pm 0.1$ ps.

“The reflectivities R for the simulations referred in Figs. 3 and 4 are observed around the transition time $t_s \pm \Delta t$ where $\Delta t = 0.1$ ps which corresponds to the scale of the ion response time $2\pi\omega_{pi}^{-1}$.” (lines 26-29 on page 7).

As the reviewer mentioned, although the overall trends are always correct, i.e., “*all the simulation points are below the maximum HB limit density*” (lines 1-3 on page 5), some of the simulation points differ from the theoretical lines in Fig. 3. In the revised manuscript, we have noted the range of the difference of R explicitly as

“The maximum density obtained in each simulation agrees with the theoretically-obtained limit density n_s for R with a difference limited in the range of ± 0.1 , ...” (lines 3-5 on page 5).

Finally, as the reviewer pointed out, it is not easy to determine t_s by only seeing the motion of the γ_c , e.g., the dashed line in Fig.4 (b). However, we can see that the plasma starts to blowout with the near critical density colored in light-blue at $t = 1.8$ ps in Fig. 4 (b), so that, we determined t_s as it is. In the other simulations, we also checked the motion of the γ_c together with the plasma blowout in determining t_s . In order to show the starting of the blowout clearly, we have added an arrow and a word “*Blowout*” in Fig. 4 (b).

Comment 4:

‘Finally, there is no 1D result at a normalized laser amplitude of 6. Is this because it did not agree with the theory? If not, this is good to know.’

Thank you for the comment. We have not executed the 1D simulation for $a_0 = 6$ just because the limit density will be higher than the presented axis of Fig. 3, i.e., up to $n_s / n_c = 150$. Namely, according to the theory, the limit density will be $n_s / n_c = 8Ra_0^2 = 288 R$. Therefore, we tested the theory at $a_0 = 6$ by a 2D simulation where the reflectivity R tends to be lower than that in 1D, so that the theory can be confirmed without putting very dense target like $>300 n_c$ in the simulation.

Comment 5:

‘This last point is related to the second most important shortcoming of the paper, which is that the bounds of applicability are not clearly explained. Throughout the paper, certain approximations are made, and there is some discussion of the density regime in which HB is ideal, but how high can the laser amplitude go before the results break down? There is some discussion of this after Eq. (4), and perhaps just making this argument clear will help. For example, I do not understand this sentence, due to grammar: “Consequently, laser lights proceed higher density than the 1D case at the same reflectivity of $R < 1$.” Throughout the paper, assumptions are made, but it is not clear that the authors sufficiently state under what conditions this is satisfied for their results to hold. It is certainly satisfied in the regime that they do their simulations that are presented. But the region of applicability needs to be explicitly stated. Experimentalists will look at the results here, plug in their values of 400 n_cr, a 2D spot, and wonder why they get crazy results. Therefore, I recommend that the limits of applicability be more explicitly stated.’

Thank you for the insightful comment. We agree with the reviewer that the discussion on the limit of the applicability of the theory is important.

The lower limit of the theory is given by $a_0 > 1$. The situation we consider here is that the laser field comes into the plasma up to the relativistic cutoff density n_c where $\gamma > 1$, and pushes electrons by the $\mathbf{v} \times \mathbf{B}$ force (the Lorentz force). This assumption of the relativistic condition $a_0 > 1$ is used in the manuscript as the approximation for the hot electron velocity, $\beta_h (= v_h/c) \sim 1$, which is written after Eq. (2) (line 65 on page 4). Note that the approximation for the plasma pressure, i.e., $n_e T_e \sim n_h T_h$ neglecting the bulk electron pressure, is valid for the relativistic cases $\beta_h \sim 1$, which can be confirmed by the current conservation relation $n_h v_h = n_b v_b$, where the bulk electrons of density n_b and flow velocity v_b contribute as the return current. Namely, $n_b T_b \propto n_b v_b^2 = n_h v_h v_b \ll n_h T_h$ where the second equality is from the current conservation.

Our theory is therefore applicable in the relativistic regime where the incident laser field amplitude is greater than unity. We have added the description regarding the lower limit in lines 63-65 on page 4 as

“This approximation is valid for the relativistic condition, $a_0 > 1$ and thus $\beta_h \sim 1$ ($>> \beta_c$).”

The upper limit of the theory will be determined by whether the energy loss by the radiation reaction becomes dominant. When the laser amplitude enters the radiation dominant regime, i.e., $a_0 > 300$ typically, the pressure balance will be changed from the present theory due to the radiative energy loss. However, we note that before encounter the upper limit by the radiation loss, the theoretical limit density n_s will be much higher than plasma densities typically considered for the laser-plasma interaction: For instance, even for the laser amplitude below the radiation dominant regime $a_0 = 200$, we have $n_s = 8Ra_0^2 n_c = 3.2 \times 10^5 R n_c$. In such a case, the laser pulse will just punch through even solid density plasmas by the HB without reaching to the HB stationary state.

In the revised manuscript, we have added the following description regarding the upper limit by the radiation loss:

“When the laser field amplitude becomes super relativistic ($a_0 \gtrsim 300$), other energy loss mechanisms such as the radiation damping will come into play, and hence, the pressure balance at the LP interaction surface will change [50].” (lines 65-69 on page 4),

with referencing

[50] Pandit, R. R., Ackad, E., d'Humieres, E. & Sentoku, Y. Ponderomotive scaling in the radiative damping regime. *Phys. Plasmas* **24**, 103302 (2017).

As for the concern for the dimension, our theory covers 1D, 2D, and 3D geometries, since we introduced the geometrical factor α ($= 1,2,3$), which can represent the radial energy diffusion. We thank the reviewer for pointing out that the sentence “*Consequently, laser lights proceed higher density than the 1D case at the same reflectivity of $R < 1$.*” is not understandable. This sentence denotes that the HB limit density n_s for multi-dimensional geometry ($\alpha = 2$ or 3) obtained from Eq. (3) is higher than n_s for 1D geometry ($\alpha = 1$), if the reflectivity R is same for both geometries. We have revised the sentence in lines 93-98 on page 4 as

“In the case of multi-dimensional interactions ($\alpha > 1$), due to the additional freedom of lateral energy diffusion, the plasma pressure becomes lower even with the same R as seen from Eq. (2). Consequently, laser lights can proceed higher density in multi-dimensional geometry than in the 1D case.”

Comment 6:

‘It would also have been nice to see the actual reflectivity for the simulations. It appears, from the discussion and figures in the article, that it is a simple number, and that is the end of the story. However, I can easily imagine the reflectivity varying wildly over the time of incidence. Does this contribute to the uncertainty in the values of reflectivity that we see plotted on the figures?’

We agree with the reviewer that it is informative to show the actual reflectivity in the simulations. We have added the time evolution of the reflectivity R in the 1D PIC simulation for $a_0 = 2$, which is the same plot with Fig. A (a) in this Reply, in Fig. 2 (b) in the manuscript. We believe that by showing this example, the readers can understand how the reflectivity changes typically during the interaction.

We have also added the following description in the discussions for Fig. 3 with referencing the newly-added Fig. 2 (b):

“Although the reflectivity R changes in time, the variation of R in the shaded interval in Fig. 2 (b), i.e., ± 0.1 ps, around the HB stationary state is small about 0.01 as shown in Fig. 3 for each simulation.”
(lines 12-15 on page 5).

Comment 7:

‘In summary, the conclusions about the density limit and the onset of a blowout regime at the end of hole boring in the manuscript are original, in that although things like hole boring have been investigated in the past, none have taken them to their logical conclusion. In other words, when the simple formulae for hole boring break down, what happens at the laser-plasma interface in the long term has not previously been addressed in as much detail as this paper does. This paper puts forth not only simulations that show what happens, but a predictive formula (for a set of laser-plasma parameters) that describes the long-time behavior. The paper will certainly influence the thinking on this specific “saturation” mechanism of hole boring and what densities are achievable (for these particular parameters) and should stimulate thinking about what happens when more complexity is introduced into the problem, like 3-D motion, high density targets where the bulk electron pressure cannot be neglected, radiation losses, and so on. As discussed above, one weakness is that it appears that each simulation has an exact value of reflectivity, and therefore the actual fluctuations in the PIC

simulation values is not quantified. However, the trends certainly agree with the trends predicted by the analytic theory. It is satisfying that very specific predictions are made and that those predictions are testable by both experiment and simulation.'

Thank you for the overall evaluation. To resolve your concern regarding the reflectivity R , we have added information of R in the manuscript, so that the readers can see how the R was determined and how sensitive in time it is at around the stationary state. The bounds of applicability of the theory have been also clarified. By these revisions, the manuscript definitely overcomes the weakness the reviewer pointed out. We again thank you for the insightful comments and helpful suggestions on our paper.

Reply to Reviewer #2

Manuscript Number: NCOMMS-17-20955-T

Title: Plasma density limits for hole boring by intense laser pulses

Authors: N. Iwata, S. Kojima, Y. Sentoku, M. Hata and K. Mima

Dear Reviewer #2:

We wish to express our appreciation to the reviewer for valuable comments and insightful suggestions, which have helped us improve the paper. The following is a point-by-point response to the reviewer's comments.

Note that the reviewer's comments are shown in the box, sentences from the manuscript are written in *italic* and enclosed in double quotation marks “ ”. The revised parts are colored in **red**.

Comment 1:

‘Hole boring by the laser beam pressure has become a fashionable scheme for inertial fusion in connection with cone guided fast ignition at comparatively moderate laser intensities (see M. Tabak). For low coronal densities, far too low for fusion applications, hole boring has been shown to work (A. Pukhov and J. Meyer-ter-Vehn). Later, about one and half decade ago, numerical studies have shown that at laser intensities below 10^{20} W/cm² hole boring was not a realistic scheme because it stopped at low density of the compressed pellet; Tabak's predictions were too optimistic. However, again a decade later new studies at some 10^{22} W/cm² made fusion with hole boring again possible, not to say fashionable, with sub-ps pulses (N. Naumova, T. Schlegel et al., 2009).

On such a background the present paper has to be considered when its authors claim that hole boring with ps pulse length comes to a rest at a limiting plasma density. Their claim is founded on analytics and on 1D and 2D PIC simulations.’

We appreciate the reviewer for mentioning a clear relation of our work with the history of the hole boring study.

Comment 2:

‘I have no difficulty to verify the fundamental relations (1) – (3) from which the assertion follows, except the assumption $I/c = \dots$ after eq. (3) may be a bit optimistic, although it does not appear unreasonable. How sensitive is the result on this assumption?’

Thank you for the comment. The relation after Eq. (3), $I/c = n_c m_e c^2 \varepsilon^2 a_0^2 / 2$ is an exact relation. The derivation is shown as follows:

The relation of the intensity I and amplitude of the electric field E for the electromagnetic wave is given by $I = c \varepsilon^2 E^2 / (8\pi)$. On the other hand, the definition of the normalized laser field amplitude is $a_0 = eE / (m_e c \omega_L)$. By using these relations, intensity I is described by a_0 as $I = c \varepsilon^2 a_0^2 / (8\pi) \times (\omega_L m_e c / e)^2$ with the cutoff density, $n_c = m_e \omega_L / (4\pi e^2)$, we obtain $I/c = n_c m_e c^2 \varepsilon^2 a_0^2 / 2$, which is the relation we use for Eq. (3).

Comment 3:

‘There is another point which in the referee’s opinion needs real clarification. Have a look at Fig. 2(a) and (b) of N. Naumova, T. Schlegel *et al.*, 2009, with piston velocities for a_circular from 20 to 300: No stopping or recession is discovered! This seems to be in contradiction to the claim of the present paper. This point should be clarified.’

Thank you for your comment on the comparison with the result in N. Naumova, T. Schlegel *et al.*, Phys. Rev. Lett. **102** (2009) 025002 (Ref. [29] in the manuscript). We consider that our result has no contradiction with the paper by Naumova *et al.*

In the paper by Naumova *et al.*, they consider circularly polarized laser, which corresponds to $\epsilon^2 = 2$ in our theory, with the normalized amplitude $a_0 = 20$ to 300. We note that even for the lower amplitude $a_0 = 20$, the HB limit density becomes $n_s = 8R(\epsilon a_0)^2 n_c = 6400 R n_c$. In Fig. 2 (b) in Naumova’s paper, the plasma density is considered only up to $n_i = 600 n_c$. If the reflectivity is lower than 10%, the HB limit density becomes $\sim 600 n_c$, however, the reflectivity in their case is expected to be higher than 10% since they discuss 1D condition mainly. Therefore, the stationary state does not appear in the region considered in her paper.

Fig. 2 (a) and (b) in N. Naumova *et al.*, Phys. Rev. Lett. **102** (2009) 025002.

Comment 4:

‘A final remark. It has become a custom to reference “Wilks scaling” in connection with energetic electron generation. There exist numerous other scaling laws, all of them of doubtful value. However, among them, Wilks’ scaling is the worst.’

Thank you for the comment. In the manuscript, we mentioned the Wilks scaling as one of the most referenced example of the scaling law for the hot electron temperature. According to the reviewer’s comment, we have added another frequently-referenced empirical scaling law $T_h \sim (I\lambda^2)^{1/3}$, i.e., so called Beg scaling, in Fig. 2 (c) (green dotted line). Correspondingly, we have added the description for the Beg scaling in line 89 on page 4 as

“... higher than the ponderomotive temperature T_p and also than the Beg scaling T_B [49].”

with referencing

[49] Beg, F. N. *et al.* A study of picosecond laser-solid interactions up to 1019 W cm⁻². *Phys. Plasmas* **4**, 447 (1997).,

and the caption of Fig. 2 as

“The slopes of the one-dimensional relativistic Maxwellian energy distribution function above 0.5 MeV are shown by the blue-dashed and green-dotted lines assuming the ponderomotive temperature at the maximum reflectivity $R = 1$, i.e., $T_p = 0.63$ MeV, and the Beg scaling $T_B = 0.44$ MeV, respectively.”

There are other scaling laws such as Puhkov’s scaling, however, such a comparison among these conventional scaling laws is not the point of this paper, and therefore, the two examples, Wilks and

Beg, would be enough to show for discussing the superthermal electron generation in Fig. 2. We appreciate the reviewer's opinion.

Comment 5:

'CONCLUSION: The paper submitted by N. Iwata et al. contains a very important statement. It will have a major impact on forthcoming studies on inertial fusion. It is presented in a clear way, the reader can follow all steps. No doubt, after clarification of the two points above the paper deserves publication in NCOMMS.'

Thank you for the supporting conclusion that our manuscript deserves publication in Nature Communications after clarifying the commented points. We believe that we have now given a clarification for each point, and hence, the revised manuscript will be suitable for publication in Nature Communications.

Reply to Reviewer #3

Manuscript Number: NCOMMS-17-20955-T

Title: Plasma density limits for hole boring by intense laser pulses

Authors: N. Iwata, S. Kojima, Y. Sentoku, M. Hata and K. Mima

Dear Reviewer #3:

We wish to express our appreciation to the reviewer for valuable comments and insightful questions, which have helped us improve the paper. The following is a point-by-point response to the reviewer's comments.

Note that the reviewer's comments are shown in the box, sentences from the manuscript are written in *italic* and enclosed in double quotation marks “ ”. The revised parts are colored in **red**.

Comment 1:

‘The manuscript by Iwata et al. entitled "Plasma density limits for hole boring by intense laser pulses" describes a simple and elegant theoretical estimate of the maximum electron density possibly reached by laser hole boring, depending on the laser intensity and the plasma reflectivity. The authors back up their approach with PIC simulation results, approximately agreeing with their theoretical predictions. Except from a few typos and unclear sentences, the manuscript quality is good, and their approximations seem justifiable (although not always explicitly justified in the manuscript).’

We greatly appreciate the evaluation especially highlighting the theory of the maximum electron density possibly reached by the hole boring. In the following in this Reply, we make the suggested ambiguous/confusing points clear.

Comment 2:

‘I believe that the current manuscript does not provide sufficient insight into what has substantially changed from previous theories. It appears that no other authors had previously attempted to estimate this maximum density, even though there have been more complete and detailed theories of hole boring. It is difficult to judge the article's novelty without a better comparison. A hint of explanation seems to be given at line 16, saying that previous approaches were "based on the momentum transfer equation from laser to ions", causing no density limit. Later, at lines 31 and 85, they claim to use a pressure balance equation, and invoke the electron fluid equation of motion at line 95, that they unfortunately did not detail nor reference. What has changed compared to the previous approaches? My understanding is that the novelty lies in the electron pressure term that has been introduced, revealing that the momentum flux eventually leaks into hot electrons, preventing hole boring to continue. This aspect has not been made clear.’

We appreciate the comment. There are two important changes from the previous theories:

One is to bring a new aspect of the HB that the *stationary state* is established in the HB process eventually where negative and positive ion density flows cancel each other, so that the ion momentum flux term does not appear in the pressure balance equation Eq. (1), as the reviewer expected. This aspect of the formation of the stationary state has never been considered in the previous studies. Here, a key to reach to the stationary state is the existence of the negative ion density flow, which is driven by the hot electron cloud behind the HB surface. In order to clarify this point, we have added the

following descriptions in the discussion for Fig. 1:

“The establishment of the stationary state is owing to the electron heating during the HB, by which a hot dense electron cloud is formed behind the HB surface, and thus the negative ion density flow is driven.” (lines 19-23 on page 3),

and

“Note that the stationary state never appears in the conventional aspect of the HB where the electron heating is not taken into account.” (lines 27-29 on page 3).

Another important change from the previous studies is to introduce the electrostatic potential, which corresponds to the surface tension, on the RHS of Eq. (1). Note that regarding the electron pressure term, previous studies (Refs. [28] and [30] in the manuscript) have added the electron pressure to the conventional momentum transfer equation which has the ion momentum flux term as $(1+R) I/c = n_e T_e + 2M_i n_i v_f^2$. They mentioned that due to the inclusion of the electron pressure term, the theoretical estimation for the hole boring speed is decreased. However, since they do not consider the stationary state, which we newly suggested here, the concept of the density limit of the hole boring never appears in the previous studies. In order to clarify this point, we have added the following descriptions after Eq. (1):

“Note that some of the previous papers have also included the electron pressure term [28, 30]. However, the stationary state sustained by the surface tension of the field has never been considered, which is the critical difference of Eq. (1) from the conventional descriptions of the HB.” (lines 41-46 on page 4).

Regarding Eq. (1), we have also added the derivation in the **Methods** section as the subsection **“Derivation of the pressure valance equation”**. In the final paragraph of this sub section, we have described the relation between Eq. (1) (Eq. (10) in the **Methods** section) and the momentum transfer equation Eq. (5) which is used for the HB stage conventionally as

“Note that in the HB stage where the ion front has not reached the stationary state, the charge separation field eventually moves the ion front in the forward (positive-x) direction. Therefore, for describing the HB stage, the last term on the RHS of Eq. (10) is replaced by the kinetic energy density of ions moving with the HB velocity v_f , and then the momentum transfer equation in the frame moving with the velocity v_f is obtained as Eq. (5).”

We believe that the above descriptions added in the revised manuscript can state the advantage of our theory clearly.

Comment 3:

‘A second, more important point, that the authors have not made, is why their elegant formula would impact any of the applications they have listed. The participation of hot electrons in the momentum conservation is certainly an interesting point to make, but its impact is not obvious. I believe that, for publication in Nature Communications, the authors should put their result in the perspective of its outcomes in related studies. Some justification attempts are given in the introduction. For instance, line 6, "energy absorption and momentum transfer ... are fundamental issues". But how did this article

address this problem? Another example, line 26, "the plasma blowout to the front side is essential to be figured out". Again, I don't see how this article has brought new insight here. None of the cited applications (channeling, high harmonics, plasma mirrors) are directly examined or even commented in the light of the authors findings. In its current form, I cannot see any "critical" advance (requested by Nature Communications) in the understanding of the hole-boring theory or in its applications. Among the proposed applications, I only see fast-ignition benefiting from this result, as the maximum density plays an important role in its design; the authors also provide a way to decide the laser optimal duration, which is handy. However, the formula only attempts at providing estimates, made even more uncertain due to the unknown laser reflectivity. This hardly provides a critical advance in fast-ignition design.'

Thank you for the comment for the perspective to advance the applications. As the reviewer intended in the beginning of **Comment 1**, the aim of this paper is to provide the simple but *fundamental* formula which presents an essential nature of the laser plasma interaction. For this reason, we do not go further to discuss actual applications, and limit discussions in showing the superthermal electron generation in the blowout phase as a fundamental outcome of the density limit theory. However, we agree with the reviewer on the point that the readers will want to know the perspective of its outcomes in the related studies. In the below, we revisit our aspect for the impact of the findings in this paper, and explain the revisions of the manuscript we have made in accordance with to the reviewer's comment.

For the applications of the HB that we cited in the introduction, i.e., laser channeling, high harmonic generation, and plasma mirror, the HB plays an important role in steepening the plasma surface which determines the efficiency of these applications. How long the steepened clean interface is sustained is an essential question for these applications, which could be answered by our theory. In order to make this point clear, we have changed the description for these applications in the introduction as

"The HB or the surface steepening has been considered to be important for applications, e.g., laser channeling [32-34], high harmonic generation [35, 36], and plasma mirror [37, 38]. For these applications, how long the steepened clean interface is sustained is an essential question." (lines 26-31 on page 1),

and added sentences explaining how our theory impacts these applications in the **Conclusion** as

"We derived the transition time t_s , at which the plasma turns from the HB to the blowout, as Eq. (8). The transition time t_s is critical for applications such as laser channeling, high harmonic generation, and plasma mirror, which require steepened clean interface for efficient operation." (lines 30-36 on page 6).

On the other hand, the plasma blowout accompanying superthermal electrons in the multi-ps interaction will be beneficial for applications: energetic ion acceleration, pair-plasma generation, etc. Actually, as we have written in the introduction (lines 39-58 on page 1), in recent LFEX experiments, such superthermal electron generations have been observed [Yogo (2017) (Ref. [46] in the manuscript), [42] Kojima (2016)], and an efficient ion acceleration owing to the high energy electrons generated by the multi-ps laser interaction was reported [[46] Yogo (2017), [47] Iwata (2017)]. Recently, the NIF-ARC experiment also demonstrated the superthermal electron generation in the few-tens of ps pulse interaction [S. C. Wilks *et al.*, APS-DPP Meeting (GO5.00015), Oct. 24, 2017] They presented the resulting high energy ion acceleration, however, the generation mechanism of such superthermal electrons is still under discussion. Our work can provide an insight for these experimental results.

In order to show the above critical advance of our work clearly, we have added the example of these applications of multi-ps lasers explicitly in the last sentence in the **Conclusion** as

*“This blowout plasma interacts with the laser field resulting **copious superthermal electrons** which **are important to increase the efficiency of the applications, e.g., ion acceleration [46, 47] and pair-plasma creation [51, 52] with multi-ps kilo-Joule laser lights.**”*

with adding the following references:

[51] Chen, H. *et al.* Relativistic positron creation using ultraintense short pulse lasers. *Phys. Rev. Lett.* **102**, 105001 (2009).

[52] Sarri, G. *et al.* Generation of neutral and high-density electron-positron pair plasmas in the laboratory. *Nat. Commun.* 6:6747 doi: 10.1038/ncomms7747 (2015).

As the reviewer pointed out, the reflectivity R is an unknown parameter in the present theory. However, recently, efforts have been made in observing the reflectivity in experiments, which could be one of the solution to obtain R in order to apply our theory to estimate the HB limit density n_s .

We believe that by the revisions above, the impact of our findings can now be seen clearly.

Comment 4:

‘To conclude, I find this paper of good quality and leads to elegant and handy formulae. It brings interesting insight in the momentum balance for hole boring (although this aspect should be discussed more). My main objection is that the authors have not shown why this has sufficient impact in the community. Without an explicit critical advance, which I understand is necessary for publication in Nature Communications, I would recommend for publication in a more specific journal.’

We appreciate the overall review comments. As the reviewer pointed out, the most important role we believe for this paper is to provide the simple but *fundamental* formula which presents an essential nature of the laser hole boring. By adding the description for the impact of our theory on actual applications according to the reviewer’s comment, we believe that the paper has been improved significantly to deserve publication in Nature Communications.

We also thank the following comments on detailed points. We have revised the manuscript accordingly.

Comment 5:

‘1/ line 10: To my knowledge, most people in the field employ the term "critical density" instead of "cutoff density". In other fields, one can encounter "cutoff frequency", but this seems rare in this community. I suggest a note be given to ensure universal understanding.’

Thank you for the comment. We recognize the term ‘critical density’ is used frequently. Following your comment, we have revised the word ‘cutoff’ to ‘critical’ in the revised manuscript.

Comment 6:

‘2/ line 32: The equal sign in the parentheses should be an approximately equal sign. This applies also line 174.’

We totally agree with the comment. We have revised the corresponding sentences from

“....., we derive the limit density for the HB as $8Ra_0^2 n_c (= 8RI_{18}\lambda_{\mu\text{m}}^2 n_c)$,”,
“The HB limit density is found to be $n_s = 8Ra_0^2 n_c (= 8RI_{18}\lambda_{\mu\text{m}}^2 n_c)$ ”

to those using an approximately equal sign and more precise value as

“....., we derive the limit density for the HB as $8Ra_0^2 n_c (\sim 5.8RI_{18}\lambda_{\mu\text{m}}^2)$,” (line 67 on page 1),
“The HB limit density is found to be $n_s = 8Ra_0^2 n_c (\sim 5.8RI_{18}\lambda_{\mu\text{m}}^2 n_c)$ ” (line 23 on page 6).

Comment 7:

‘3/ line 71: How is R chosen? The manuscript states "We use the reflectivity R at the transition time from the HB to the blowout". Does that mean R is considered constant and the value is somehow extracted from the simulation at a time between 0.4 and 0.6 ps? And, instead of this calculation of the relativistic critical density, why not simply measure the place where there is no more laser field?’

As the reviewer mentioned, we used the reflectivity R at the transition time from the HB to the blowout, considering that R can be regarded to be almost constant around the transition time. We agree with the reviewer’s opinion that more detailed description on the time evolution of the reflectivity R is necessary.

In this revision, we have added the variation of R within $t_s \pm \Delta t$ where $\Delta t = 0.1$ ps for each simulation points on Figs. 3 and 4. This time width Δt corresponds to the scale of the ion response time $2\pi\omega_{pi}^{-1}$, as is noted in the manuscript as “.... the ion response time scale, $2\pi\omega_{pi}^{-1}$ which is in the order of 100 fs at the critical density.” (lines 47-49 on page 2). We have also added a sentence to explain how (for which time) we obtained R in the simulations in the **Methods** section as

“The reflectivities R for the simulations referred in Figs. 3 and 4 are observed around the transition time $t_s \pm \Delta t$ where $\Delta t = 0.1$ ps which corresponds to the scale of the ion response time $2\pi\omega_{pi}^{-1}$.” (lines 26-29 on page 7)

In addition, we have added the time evolution of the reflectivity R in the 1D PIC simulation for $a_0 = 2$ in Fig. 2 (b), and put the shades in Figs. 2 (a) and (b) to indicate the transition time interval from the HB to the blowout. The description for Fig. 2 (b) has been added in lines 36-42 on page 3 as

“Figure 2 (b) shows the reflectivity R , which keeps an almost constant value around the transition time from the HB phase to the blowout phase.”

and

“....., however, during the time $t = 0.5$ ps - 0.7 ps that is shaded in Figs. 2 (a) and (b),”.

We believe that by showing Fig. 2 (b) as an example of the time evolution of R , the readers can understand how the reflectivity changes typically during the interaction, and can also see that the variation of R around $t = t_s$ is small. We have also added the following description in the discussions for Fig. 3 with referencing the newly-added Fig. 2 (b):

“Although the reflectivity R changes in time, the variation of R in the shaded interval in Fig. 2 (b), i.e.,

±0.1 ps, around the HB stationary state is small about 0.01 as shown in Fig. 3 for each simulation.” (lines 11-14 on page 5).

Regarding the method to obtain the maximum density in the simulation, it is not easy to specify the exact location of the HB front from the laser field profile. This is because the amplitude of the laser field oscillates with the rapid laser frequency and drops gradually within the scale of skindepth. Instead, we used the time-averaged density profile to remove the high frequency oscillation.

Comment 8:

‘4/ line 79: The expression "electrons with densities $n_e < 0.1 n_c$ " should be corrected. The authors certainly refer to "the electrons in the region where $n_e < 0.1 n_c$ ". This appear again line 80.’

Thank you for this suggestion. According to the comment, we have revised the previous sentence

“..., mainly electrons with densities $n_e < 0.1 n_c$ interact directly with”

as

“..., mainly *the electrons in the region where* $n_e < 0.1 n_c$ interact directly with” (lines 51-52 on page 3).

as the reviewer suggested. Regarding the expression

“..., electrons with near critical densities $0.1 n_c < n_e < n_c$ expand about $1 \mu\text{m}$,”,

we have revised it as

“..., *the region where* $0.1 n_c < n_e < n_c$, expands about $1 \mu\text{m}$,” (line 2 on page 4).

Comment 9:

‘5/ line 92: The expression of the charge separation field E_s appears wrong. I believe it is $E_s = 4 \pi e n_e l_s/2$. However, the subsequent calculations are correct.’

We appreciate the comment. The previous expression “ $2\pi e^2 n_e^2 (l_s/2)^2$ ” was not for E_s , but for $E_s^2/8\pi$. We are sorry for this typo. We have corrected the equation in the text as “ $E_s = 2\pi e n_e l_s$ ” (line 27 on page 4).

Comment 10:

‘6/ line 95: The quantity phi should be defined.’

Thank you for the comment. We have moved the details of the derivation of Eq. (1) to the subsection “**Derivation of the pressure balance equation**” in the **Methods** section. We have defined ϕ there as “*where ϕ is the electrostatic potential,*” (line 35 on page 7).

Comment 11:

‘7/ line 101: The quantity T_c should be T_e .’

In this sentence, we defined the bulk electron temperature T_c . However, T_c is never used thereafter, hence, we have removed the description “.....with temperatures T_h and T_c , respectively.”, and added only the definition of the hot electron temperature in the following sentence as

“We here assume $n_e T_e \sim n_h T_h$ where T_h is the hot electron temperature,

Comment 12:

‘8/ Figure 3: Having 2D simulations compared to 1D predictions is troubling. The authors could consider duplicating their figure for clarity: one panel for 1D and one panel for 2D.’

Thank you for the comment. The reason we plot 2D simulation results on 1D theoretical lines is that the 2D simulations we show in Fig. 3 assume a wide laser focusing, except one case shown by the white circle. Namely, as described in the **Methods** section (lines 17-25 on page 7), the spot size is $60 \mu\text{m}$ which is much larger than the plasma scale length L , and thus the interaction can be regarded as quasi-1D for the wide focus cases. This is why we plot 2D simulation results with the 1D theory lines. Note that only one white circle at $\epsilon_{a_0} = 4$ corresponds to the tight focus 2D simulation, for which we write n_s obtained using the geometrical factor $\alpha = 2$ for 2D in the plot.

We understand it was confusing without explicitly describing the above discussion on the quasi one-dimensionality of the 2D simulations. We have specified the 2D simulations as quasi-1D explicitly as follows:

“We also present the maximum density to which the laser light can reach in 1D and 2D (*quasi-1D*) PIC simulations.” (from line 105 on page 4 to line 1 on page 5).

Please note that the quasi-1D condition for the 2D simulations is also mentioned in the caption of Figs. 3 and 4. In this revision, we have further modified the captions of Figs. 3 and 4 from

“Red circles and red square are the PIC results for two-dimensional geometry with widely-focused (spot size of $60 \mu\text{m}$) linear-p and circular polarizations, respectively.”

to

“Red circles and red square are the 2D PIC results for *quasi-1D* geometry with widely-focused (spot size of $60 \mu\text{m}$) linear-p and circular polarizations, respectively.”

Comment 13:

‘9/ line 137: "Hereafter, we estimate the time scale t_s to reach the HB limit density. This time t_s corresponds to the time scale of the HB after that the plasma starts to blowout." The two cited sentences seem to contradict each other. The first sentence seems to define t_s as the time between the beginning of the interaction up to the HB density limit. The second sentence seems to define t_s as the time between the HB limit, and the appearance of a blowout plasma. I believe the second sentence is ambiguously constructed.’

We appreciate the comment. As the reviewer mentioned, what we intend to say is to define t_s as the time between the beginning of the interaction up to the HB density limit. We agree with the reviewer on the point that the second sentence could cause a confusion, and we think the first sentence is enough as the explanation for t_s . Therefore, we have removed the second sentence,

“This time t_s corresponds to the time scale of the HB after that the plasma starts to blowout.”,

from line 15 on page 5.

Comment 14:

‘10/ Fig 4a: Why is this figure showing less points than Fig 3? Did the authors verify their theory only for some of their simulations? Or is there a problem with the other simulations that are not present in Fig 4a?’

Thank you for the comment. In the simulations shown in Fig. 4, we have assumed the exponential pre-plasma distribution $n_e = \gamma n_c \exp(x/L)$ as written in line 27 on page 5, with the scale length $L = 2 \mu\text{m}$ (written in lines 51-52 on page 5). This exponential distribution corresponds to the assumption used in deriving Eq. (8). On the other hand, most of the simulations shown in Fig. 3 are different from those shown in Fig. 4, having different pre-plasma distributions. Namely, the 1D simulations shown in Fig. 3 assume a linear pre-plasma with the scale length of $1 \mu\text{m}$ as written in **Numerical simulations** in the **Methods** section as “..... with a linear pre-plasma of length $L = 1 \mu\text{m}$ whose electron density increases from zero at $x = 9 \mu\text{m}$ to the uniform plasma density $n_{e0} > 8\hat{a}_0^2$ at $x = 10 \mu\text{m}$.” (from line 63 on page 6 to line 1 on page 7). Note that the difference of the scale length L does not affect the determination of the limit density n_s . Namely, there is no dependence on L in Eq. (3).

From the above reason, we have less points in Fig. 4 than Fig. 3. In this revision, we found another simulation results assuming $L = 2 \mu\text{m}$ exponential pre-plasma, so that, we have added one blue square point (1D, C-pol.) at $\epsilon a_0 = 2$ in Fig. 4 (a).

We again thank you for the valuable comments on our paper. We kindly ask you to review the revised manuscript for publication in Nature Communications.

REVIEWERS' COMMENTS:

Reviewer #1 (Remarks to the Author):

The authors have answered all the questions I had regarding the manuscript, and have adequately addressed all my concerns. I recommend publication in this journal.

Reviewer #2 (Remarks to the Author):

Final comment of the referee on paper NCOMMS-17-20955A by N. Iwata et al.

The authors have clarified all relevant questions to my satisfaction I had raised in my previous report. Additionally, in going through the comments of referees A and C and the reply to them by the authors I appreciate the effort spent in answering the questions and removing ambiguities. I conclude with the statement from my first report:

The paper submitted by N. Iwata et al. contains a very important statement. It will have a major impact on forthcoming studies on inertial fusion.

I recommend publication of the paper in its present form.

Reviewer #3 (Remarks to the Author):

The authors of the manuscript entitled "Plasma density limits for hole boring by intense laser pulses" have answered appropriately a number of my previous remarks. I repeat that this work appears original, concise, clear and brings interesting insight into the process of hole boring.

Only one point remains difficult to assess: the impact of this work on this area of research. I stated that I did not see how the results would have significant influence on future work. The authors formulate two answers to my remark. (1) They claim that their result concerning the "transition time" will help determine the efficiency of applications because they depend on how long the hole boring can be sustained. (2) They point to the beneficial role of the plasma blowout accompanying the superthermal electrons for certain applications.

The point (2) is hardly anything new, as this plasma blowout and the superthermal electron population have been studied thoroughly already. The present manuscript does not add much in this regard. The point (1) is actually appropriate, but relates only to the last part of the manuscript, not to their "principal" result concerning the density limit. I thus consider their answer as limited.

My overall feeling is that, if these results bring novelty to the field, it will be the general idea that the superthermal electron pressure actually has importance for hole boring, and leads to stopping the process. That point might bring renewed theoretical or experimental interest, and might spark critical advances, but it is very hard to predict today.

I might not have the same perspective as the other referees on this very subjective point. I do not view this work as an objective critical advance, although it definitely has brought new insight and is well explained. Who knows: maybe future critical developments will be based on these insights, and this article might be viewed as a solid reference. However, I think that is a somewhat remote possibility.

If the elegance and concision of the results are considered as major qualities for Nature Comm., I recommend the article for publication. Furthermore, if the editors are willing to take a chance on the future impacts of this article, I also recommend it. However, if the advances made by the article must be objectively critical, I do not.